# Feedforward 4D Reconstruction for Dynamic Driving Scenes using Unposed Images

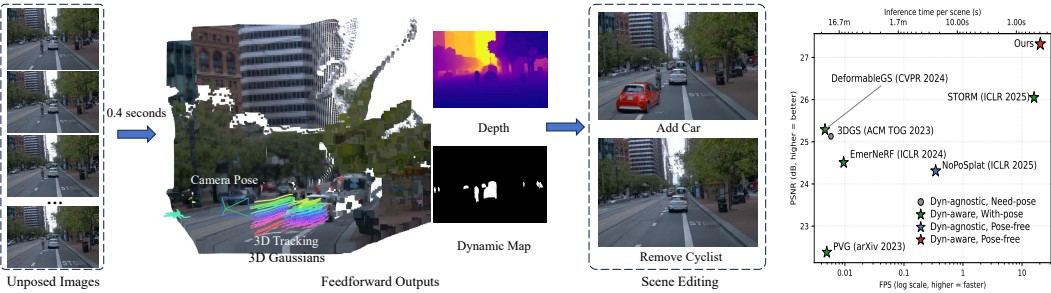

Figure 1: **Left:** Our feedforward framework reconstructs dynamic scenes from unposed images in 0.4 s, enabling editing tasks such as object addition or removal. **Right:** We achieve state-of-the-art performance with competitive speed among feedforward methods.

## ABSTRACT

Autonomous vehicles require diverse dynamic scenes for robust training and evaluation, yet existing dynamic scene reconstruction methods are often limited by slow per-scene optimization and reliance on explicit annotations or camera calibration. In this paper, we introduce a pose-free, feedforward framework for 4D scene reconstruction that jointly infers camera parameters, dynamic Gaussian representations, and 3D motion directly from sparse, unposed images. Unlike prior feedforward approaches, our model accommodates an arbitrary number of input views, enabling long-sequence modeling and improved generalization. Dynamic objects are disentangled via estimated motion and aggregated into unified 3DGS representations, while a diffusion-based refinement module mitigates flow artifacts and enhances novel view synthesis under sparse inputs. Trained on the Waymo Dataset and evaluated on nuScenes and Argoverse2, our method achieves superior performance while generalizing effectively across datasets, benefiting from the pose-free design that reduces dataset-specific biases. Additionally, the framework supports instance-level scene editing and high-fidelity view synthesis, providing a scalable foundation for real-world autonomous driving simulation.

## 1 INTRODUCTION

Autonomous vehicles navigate complex and dynamic 3D environments, requiring diverse dynamic scenes for robust training and evaluationYan et al. (2025); Wu et al. (2023); Tonderski et al. (2024); Li et al. (2025); Jin et al. (2024); Tian et al. (2023); Chen et al. (2024c); Yan et al. (2024). Recent advances in neural scene representationsShao et al. (2023); Park et al. (2021); Luiten et al. (2024); Wu et al. (2024) achieve impressive visual fidelity in reconstructing dynamic scenes from multi-timestep images. However, these approaches typically rely on per-scene optimization that minimizes photometric loss, requiring several minutes to hours to train on a single sceneMartin-Brualla et al. (2021); Zhang et al. (2024a); Kulhanek et al. (2024); Gao et al. (2024); Wang et al. (2025b). Such workflows are computationally expensive and time-consuming, making them impractical for large-scale simulation. Moreover, many existing methodsYan et al. (2024); Zhou et al. (2024) rely on explicit 3D annotations such as bounding boxes to model motion, which are labor-intensive to obtain and restrict applicability in real-world driving scenarios.

To address these limitations, STORMYang et al. (2024a) introduces a feedforward framework for fast dynamic scene reconstruction from images. While this marks an important step toward scalable 4D reconstruction, several challenges remain. First, training is limited to sequences of only four timesteps, which hinders the ability to capture long-term dynamics. Second, novel view synthesis suffers from sparse input views, often leading to noticeable fidelity degradation under large viewpoint changes. Finally, the method still assumes access to accurate camera poses, requiring an additional calibration stage that reduces its practicality for in-the-wild video sequences.

In this paper, we present a feedforward model for 4D scene reconstruction that jointly estimates camera parameters and 3D scene representation in a single pass. Compared with STORM, our framework accommodates an arbitrary number of unposed images as input, enabling the modeling of longer sequences and demonstrating improved generalization to unseen data by mitigating dataset-specific biases in camera pose. Built on a unified vision-transformer backbone, the model predicts per-frame 3D Gaussian representations and 3D motion to capture dynamic objects. Without relying on extrinsic camera calibration or instance-level annotations for dynamic objects, this framework enables efficient and scalable reconstruction of 4D scenes directly from sparse, unposed images.

To represent dynamic objects, we leverage the estimated motion and per-frame 3DGS maps to decompose the scene into static and dynamic components, and aggregate dynamic objects across frames into a unified 3DGS representation via spatial transformations. However, imperfections in motion estimation inevitably introduce artifacts into the aggregated 3DGS. To address this, we further introduce a diffusion-based refinement module that operates on rendered images, substantially improving reconstruction fidelity. Beyond artifact reduction, this refinement process mitigates the limitations of novel view synthesis under sparse-view settings, yielding higher-quality renderings from challenging viewpoints.

Trained on the Waymo Open Dataset Sun et al. (2020) and evaluated on diverse benchmarks, including nuScenes Caesar et al. (2020) and Argoverse2 Wilson et al. (2023), our method achieves superior rendering quality and generalization ability, while also improving runtime efficiency compared to optimization-based approaches. Additionally, it supports instance-level editing, as illustrated in Fig. 1. By combining high-quality 4D reconstruction with real-time performance, our framework provides a scalable and practical foundation for future autonomous driving simulation systems.

## 2 RELATED WORKS

**Dynamic scene reconstruction** aims to recover a time-varying 3D representation of a scene from an image sequence. Recent extensions of NeRF Li et al. (2022); Pumarola et al. (2021); Shao et al. (2023); Park et al. (2021); Mildenhall et al. (2021); Wang et al. (2021); Yu et al. (2022); Liu et al. (2024); Yuan & Zhao (2024) and 3DGS Luiten et al. (2024); Wu et al. (2024); Yan et al. (2024); Zhou et al. (2024); Chen et al. (2024a); Cheng et al. (2024); Zhang et al. (2024c); Ye et al. (2025) to dynamic settings have achieved high-fidelity reconstructions. Based on their motion modeling strategies, existing approaches can be broadly categorized into two main classes. The first class employs temporally conditioned representations, where time is served as an explicit input to the model. For example, PVG Chen et al. (2023) integrates periodic vibration-based temporal signals to predict 3DGS representations. However, such methods are often impractical for downstream tasks like object editing, as they lack disentangled, object-centric representations. The second class represents the scene as a compositional scene graph, where dynamic entities are modeled independently—typically as separate NeRFs Wu et al. (2023) or 3DGS Chen et al. (2024c) components. While this structure facilitates object-level manipulation, these methods often rely on external 3D annotations such as bounding boxes, which are costly to obtain. Moreover, most existing methods rely on per-scene optimization, requiring several hours to reconstruct each scene. To address these limitations, we propose a fast, feedforward dynamic reconstruction method that eliminates the need for additional annotations, enabling efficient and generalizable 4D modeling across diverse scenes.

**Feedforward reconstruction** infers 3D scene representations like NeRF or 3DGS directly from input observations via a single forward pass of a trained neural network. Unlike per-scene optimization methods, feedforward approaches are designed to generalize across diverse scenes and enable real-time inference. Several methods Hong et al. (2023); Tang et al. (2024); Wang et al. (2025a); Keetha et al. (2025) adopt vision transformers to reconstruct 3D objects from multi-view images. Subsequent works such as Flash3D Szymanowicz et al. (2024), MVSplat Chen et al. (2024b), No-

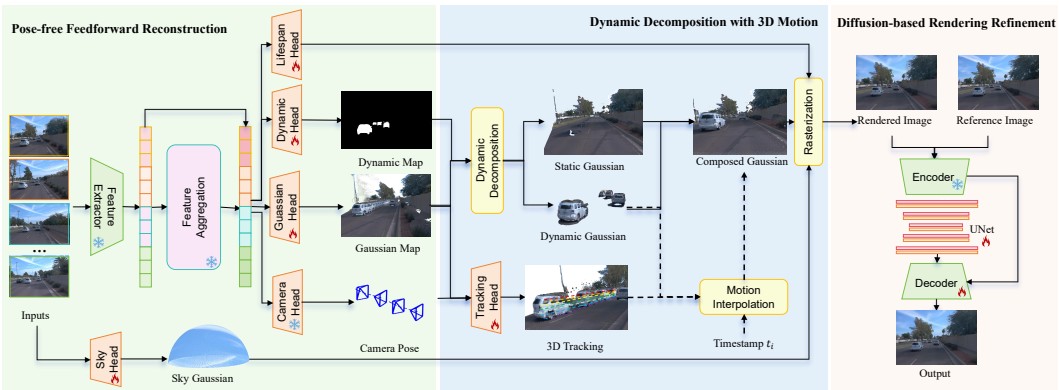

Figure 2: **Overall Architecture.** Given unposed images of dynamic scene, we estimate camera parameters, dynamic maps, and per-pixel Gaussians in a single pass. Subsequently, a motion head is employed to track dynamic objects across time, and their trajectories are interpolated to construct temporally consistent Gaussian representations. Finally, a diffusion-based rendering module refines the resulting composition, producing high-fidelity renderings.

PoSplatYe et al. (2024) and DepthSplat Xu et al. (2025) extend feedforward 3D Gaussian Splatting to the scene level, though they remain limited to static environments. L4GMRen et al. (2024) presents the first 4D reconstruction model, yet the application is mainly restricted to objects. More recently, STORM Yang et al. (2024a) introduces a feedforward framework for dynamic scenes, but it is constrained by the number of input frames and struggles with long sequences. In contrast, our method supports an arbitrary number of input frames and does not require camera poses, enabling more flexible and efficient dynamic scene reconstruction.

## 3 METHOD

In this paper, we introduce a pose-free, feedforward framework that reconstructs 3D structure directly from unposed images (Sec.3.1). To handle dynamic scenes, we incorporate a motion estimation module that predicts 3D motion and enables consistent fusion of moving objects (Sec. 3.2). Finally, a diffusion-based refinement module is further applied to enhance reconstruction quality and support high-fidelity view synthesis (Sec. 3.3). This fully feedforward design enables efficient and scalable reconstruction, delivering high-quality results directly from raw image sequences without any pose or 3D annotation as inputs. The overall architecture is depicted in Fig 2.

### 3.1 POSE-FREE FEEDFORWARD RECONSTRUCTION

Given a sequence of unposed RGB images $\{I^t \mid I^t \in \mathbb{R}^{H \times W \times 3}, t = 1, \ldots, N\}$ captured over $N$ timestamps of a dynamic scene, our objective is to reconstruct a temporally coherent 3D representation in a single forward pass, without requiring on external pose calibration or 3D annotations. To this end, we propose a feedforward model $f_\theta$ that directly maps the image sequence to a dynamic 3D scene representation. Concretely, the model jointly estimates the per-frame camera parameters $\Pi^t$ and the 3D scene representation for each timestamp.

**Scene Representation** We represent the scene at each frame using a pixel-aligned Gaussian map $G^t \in \mathbb{R}^{H \times W \times 15}$, where each pixel-aligned primitive at $(i, j)$ encodes RGB color $c_{i,j}^t \in \mathbb{R}^3$, 3D mean position $\mu_{i,j}^t \in \mathbb{R}^3$, rotation quaternion $r_{i,j}^t \in \mathbb{R}^4$, scale $s_{i,j}^t \in \mathbb{R}^3$, opacity $o_{i,j}^t \in \mathbb{R}$ , and a lifespan parameter $\sigma_{i,j}^t \in \mathbb{R}^+$, which controls its temporal influence by modulating opacity over time. Specifically, given the predicted parameters at timestamp $t$, the opacity at another timestamp $t'$ is computed as:

$$o^{t'} = o^t \cdot e^{-\frac{1}{2} \cdot \frac{(t'-t)^2}{\sigma^t}}, \tag{1}$$

$\sigma$ controls the temporal spread of the Gaussian, and a larger $\sigma$ leads to a longer-lasting Gaussian in time, while a smaller $\sigma$ results in faster temporal fading.

The first frame $I^1$ is designated as the reference frame, with its camera origin serving as the world coordinate origin. All subsequent frames are aligned to this reference, ensuring consistent 3D positioning across time.

**Sky Modeling** To account for distant background regions such as the sky, we introduce a separate set of sky Gaussians denoted by $G_{sky}$. Their centers are uniformly sampled on a hemisphere of fixed radius $r_{sky}$, chosen to approximate an effectively infinite background. Each sky Gaussian is assigned a fixed rotation and opacity. To determine their appearance, we project their 3D coordinates onto the input images and extract the corresponding pixel colors. These initial colors, along with the Gaussian scale parameters, are then refined by a lightweight MLP $\mathcal{H}_{sky}$ to improve consistency and realism in sky modeling.

**Model Architecture** Following VGGT Wang et al. (2025a), we adopt a 24-layer ViT architecture as the backbone of our feedforward model $f_\theta$. The input images are first partitioned into patches and transformed into token sequences, which are subsequently encoded by a DINO-pretrained feature extractor Zhang et al. (2022) to obtain rich visual representations, denoted as $F_{\text{dino}}$. These features are then refined through an alternating-attention mechanism, yielding $F_{\text{attn}}$. The resulting features are processed by multiple prediction heads: the camera head $\mathcal{H}_{\text{cam}}$ for estimating camera poses, the Gaussian head $\mathcal{H}_{\text{gs}}$ for generating 3D Gaussians, and the lifespan head $\mathcal{H}_{\text{life}}$ for lifespan parameters.

During training, we leverage pretrained priors by freezing the feature extractor and camera head, while training the remaining heads from scratch. Feeding $F_{\text{attn}}$ into the prediction heads, the camera pose is obtained as $\Pi^t = \mathcal{H}_{\text{cam}}(F_{\text{attn}})$. However, we observe that $F_{\text{attn}}$ primarily encodes high-level semantics and lacks sufficient detail for appearance reconstruction. To mitigate this issue, we fuse $F_{\text{attn}}$ with the original DINO features, thereby enhancing spatial fidelity. The fused feature is then used by the Gaussian head to generate the Gaussian splatting map as $G^t = \mathcal{H}_{\text{gs}}(F_{\text{dino}}, F_{\text{attn}})$.

This unified framework enables reconstruction of the 3D scene structure across all timestamps. Nevertheless, in dynamic environments, naïvely aggregating Gaussian maps across time leads to incoherent reconstructions due to object motion. To address this, we introduce a motion field that explicitly models dynamic object trajectories, as detailed in the following section.

## 3.2 Dynamic Decomposition with 3D Motion

Given the predicted Gaussian maps, directly aggregating them over time would produce ghosting artifacts due to moving objects, as illustrated in Fig. 2 (Gaussian map). Therefore, we extend the feedforward model with a dynamic head $\mathcal{H}_{\text{dy}}$ that predicts the probability of dynamic regions, formulated as $M_d^t = \mathcal{H}_{\text{dy}}(F_{\text{attn}})$. This dynamic map distinguishes moving objects from the static background, enabling us to decompose each Gaussian map $G^t$ into a static component $G_s^t$ and a dynamic component $G_d^t$:

$$G_s^t = G^t \odot (1 - M_d^t), \quad G_d^t = G^t \odot M_d^t, \tag{2}$$

where $\odot$ denotes element-wise multiplication. For each timestamp $t$, the full Gaussian representation is constructed by combining the sky Gaussian, the static components from all frames, and the dynamic component of the current frame as:

$$\hat{G}^t = \left( \bigcup_{t'=1}^{N} G_s^{t'} \right) \cup G_d^t \cup G_{\text{sky}}. \tag{3}$$

here $G_d^t$ denotes the dynamic Gaussians at frame $t$, and $\bigcup_{t'=1}^{N} G_s^{t'}$ represents the union of static Gaussians from all frames. The rendered image at timestamp $t$ is then obtained via the differentiable renderer, defined as:

$$\hat{I}^t = Renderer(\hat{G}^t, \Pi^t), \tag{4}$$

where $\Pi^t$ denotes the predicted camera parameters.

**3D Motion Estimation** In practice, the input timestamps are sparse. For an intermediate time $t_i$ with $t_i \notin \{1, \ldots, N\}$, corresponding dynamic Gaussian representation is not directly available. To address this, we explicitly model object motion over time. We exploit correspondences between

input images and introduce a motion head that predicts 3D motion for a set of query pixels $\mathcal{Q} \in \mathbb{R}^{q \times 2}$, where $q$ is the number of queries. Concretely, for any pair of timestamps $t_a, t_b \in \{1, \ldots, N\}$, the motion head estimates 3D motion $F(t_a, t_b) \in \mathbb{R}^{q \times 3}$, which provides per-pixel 3D displacement vectors, enabling temporal alignment of Gaussian primitives across frames.

Specifically, we introduce a transformer-based motion head $\mathcal{H}_{\mathrm{motion}}$, which jointly processes 2D images and 3D points extracted from Gaussian maps. The images are first encoded into multi-scale features, which are then associated with 3D points to construct a spatio-temporal feature cloud. For each timestamp $t_a$, we select the one-valued pixels in the dynamic map $M_d^{t_a}$ as query pixels $\mathcal{Q}$, back-project them to initialize their 3D positions, and iteratively refine their trajectories via neighborhood-to-neighborhood attention. Formally, the motion head is defined as:

$$F(t_a, t_b) = \mathcal{H}_{\mathrm{motion}}(\mathcal{Q} \mid G^{t_a}, G^{t_b}, I^{t_a}, I^{t_b}), \tag{5}$$

where $G^{t_a}$ and $G^{t_b}$ denote the Gaussian maps and $I^{t_a}$, $I^{t_b}$ the corresponding images at times $t_a$ and $t_b$. The motion head is initialized with pretrained weights Zhang et al. (2025) and subsequently finetuned using a photometric loss on interpolated frames.

**Motion Interpolation** With the 3D motion, we can estimate the dynamic Gaussian representation at an intermediate time $t_i \in [t_a, t_b]$, where $t_a$ and $t_b$ are adjacent timestamps. Specifically, we interpolate the mean coordinates $\mu_d^t$ of the dynamic Gaussians using the motion prediction as:

$$\mu_d^{t_i} = \mu_d^{t_a} + \omega_{t_i} \cdot F(t_a, t_b), \quad \omega^{t_i} = \frac{t_i - t_a}{t_b - t_a}, \tag{6}$$

where $\omega^{t_i}$ is a linear interpolation weight and $M_d^{t_a}$ denotes the dynamic mask at $t_a$. This yields the dynamic Gaussians $G_d^{t_i}$ at the intermediate timestamp $t_i$, which are then combined as Eq. 3 to obtain the full Gaussian representation. For camera pose $\Pi^{t_i}$ at timestamp $t_i$, we linearly interpolated the translation between $\Pi^{t_a}$ and $\Pi^{t_b}$, while the rotation was interpolated using spherical linear interpolation (SLERP) on quaternions.

Notably, our model predicts all camera parameters, Gaussian representations, 3D motion, and dynamic masks in a single forward pass, enabling simultaneous 4D reconstruction and scene understanding. The full forward process is defined as:

$$G_{\mathrm{sky}}, \{G^t, F^t, M_d^t, \Pi^t\}_{t=1}^N = f_\theta(\{I^t\}_{t=1}^N), \tag{7}$$

where $G_{\mathrm{sky}}, G^t, F^t, M_d^t, \Pi^t$ are the sky Gaussian, Gaussian map, 3D motion, dynamic map and camera pose respectively, $f_\theta$ denotes the feedforward network, and $\{I^t\}_{t=1}^N$ is the set of input images.

**Training Objectives** We train the feedforward reconstruction model including the motion estimation head, in an end-to-end manner. For each training iteration, we randomly sample $N \in [4, 8]$ input images and generate $2N$ images by interpolating between the sampled frames. A reconstruction loss is applied to all $2N$ images, combining an $\ell_2$ term with a perceptual LPIPS term:

$$\mathcal{L}_{\mathrm{rgb}} = \mathcal{L}_{\ell_2} + \lambda_{\mathrm{LPIPS}} \mathcal{L}_{\mathrm{LPIPS}}. \tag{8}$$

To supervise opacity and dynamic maps, we adopt binary cross-entropy losses as:

$$\mathcal{L}_{\mathrm{opacity}} = \mathrm{BCE}(M_{\mathrm{sky}}, \hat{M}_{\mathrm{sky}}), \quad \mathcal{L}_{\mathrm{dynamic}} = \mathrm{BCE}(M_d, \hat{M}_{\mathrm{dynamic}}), \tag{9}$$

where $M_{\mathrm{sky}}$ is derived from the rendered opacity with ground-truth $\hat{M}_{\mathrm{sky}}$, and $M_d$ denotes the predicted dynamic map with ground-truth $\hat{M}_{\mathrm{dynamic}}$. We impose an $\ell_1$ regularization for lifespan parameters under the assumption that most of the scene is static as $\mathcal{L}_{\mathrm{lifespan}} = \left| \frac{1}{\sigma} \right|_1$.

The overall training objective is a weighted combination of these components:

$$\mathcal{L}_{\mathrm{feedforward}} = \mathcal{L}_{\mathrm{rgb}} + \lambda_{\mathrm{opacity}} \mathcal{L}_{\mathrm{opacity}} + \lambda_{\mathrm{dynamic}} \mathcal{L}_{\mathrm{dynamic}} + \lambda_{\mathrm{lifespan}} \mathcal{L}_{\mathrm{lifespan}}. \tag{10}$$

### 3.3 DIFFUSION-BASED RENDERING REFINEMENT

Although the estimated motion field provides a plausible description of object dynamics, interpolation still produces artifacts such as ghosting and disocclusion gaps, due to deviations in the estimated

| Method | Render Quality | | | Inference time (s) | Dynamic | Pose-free |
|---|---|---|---|---|---|---|
| | PSNR ↑ | SSIM ↑ | D-RMSE ↓ | | | |
| EmerNeRFYang et al. (2023) | 24.51 | 0.738 | 33.99 | 14min | ✓ | ✗ |
| 3DGSKerbl et al. (2023) | 25.13 | 0.741 | 19.68 | 23min | ✗ | ✗ |
| PVGChen et al. (2023) | 22.38 | 0.661 | 13.01 | 27min | ✓ | ✗ |
| DeformableGSYang et al. (2024b) | 25.29 | 0.761 | 14.79 | 29min | ✓ | ✗ |
| LGMTang et al. (2024) | 18.53 | 0.447 | 9.07 | 0.06s | ✗ | ✗ |
| GS-LRMZhang et al. (2024b) | 25.18 | 0.753 | 7.94 | **0.02s** | ✗ | ✗ |
| MVSplat Chen et al. (2024b) | 20.56 | 0.697 | 10.13 | 0.08s | ✗ | ✗ |
| NoPoSplat Ye et al. (2024) | 24.31 | 0.751 | 9.08 | 23.22s | ✗ | ✓ |
| DepthSplat Xu et al. (2025) | 23.26 | 0.696 | 10.05 | 0.11s | ✗ | ✗ |
| STORM* Yang et al. (2024a) | 26.05 | 0.819 | 5.91 | 0.50s | ✓ | ✗ |
| STORMYang et al. (2024a) | 26.38 | 0.794 | **5.48** | 0.18s | ✓ | ✗ |
| Ours | **27.41** | **0.846** | 5.56 | 0.39s | ✓ | ✓ |

Table 1: **Quantitative comparison on the Waymo dataset**. Higher PSNR and SSIM, and lower D-RMSE indicate better performance.(* denotes the results from our replication)

motion and the limited robustness of the 3DGS representation under large rotations and translations. To enhance realism, we further incorporate an image-space refinement stage with diffusion model. This post-render module serves two key purposes: (1) suppressing interpolation artifacts to deliver high-quality renderings, and (2) compensating for novel view gaps caused by the limited field of view in the input observations.

We build our refinement model $f_{\text{diffusion}}$ on a single-step diffusion framework Wu et al. (2025), consisting of a frozen VAE encoder, a UNet denoiser, and a LoRA fine-tuned decoder. As illustrated in Fig. 2, given a rendered image $\hat{I}^{t_i}$ and a reference image $I_{\text{ref}}$ randomly sampled from the input sequence, we first concatenate them frame-wisely and encode the result into the latent space using the VAE encoder. The latent embedding is subsequently processed by the UNet denoiser and decoded to yield the refined image $\tilde{I}^{t_i}$. Formally, the process is formulated as:

$$\tilde{I}^{t_i} = f_{\text{diffusion}}(\hat{I}^{t_i}, I_{\text{ref}}). \tag{11}$$

The diffusion model is also supervised under an $\ell_2$ reconstruction loss between the model output $\tilde{I}^{t_i}$ and the corresponding ground-truth image $I^{t_i}$, complemented by a perceptual LPIPS loss. To further enhance sharpness and fine details, we also incorporate a style loss based on the Gram matrices of VGG-16 features. The overall objective for fine-tuning the model is defined as

$$\mathcal{L}_{\text{diffusion}} = \mathcal{L}_{\text{Recon}} + \mathcal{L}_{\text{LPIPS}} + \lambda_{\text{Gram}}\mathcal{L}_{\text{Gram}}. \tag{12}$$

## 4 EXPERIMENT

### 4.1 EXPERIMENTAL PROTOCOL

**Datasets** We conduct experiments on the Waymo Open DatasetSun et al. (2020), which contains real-world autonomous driving logs. For training, we use the full Waymo Open Dataset training split, which comprises 798 scenes, each containing 190–200 frames. For evaluation, we use the Waymo Open Dataset test split, which comprises 202 scenes and covers challenging scenarios such as multi-vehicle interactions, nighttime driving, and rainy weather. To assess the generalization ability of our method across varied driving conditions, we evaluate on two additional benchmarks: nuScenes Caesar et al. (2020) and Argoverse2 Wilson et al. (2023). For quantitative evaluation on 4D reconstruction, performance is reported using PSNR and SSIM for images, as well as root mean square error (RMSE) for depth estimation. For 3D tracking evaluation, we adopt End-Point Error in 3D (EPE3D), Acc5, Acc10, and angular error as metrics, following Yang et al. (2024a).

### 4.2 COMPARISON WITH SOTA METHODS

**Baselines** We compare our approach with a broad set of reconstruction methods, including optimization-based techniques such as EmerNeRF Yang et al. (2023), 3DGS Kerbl et al. (2023),

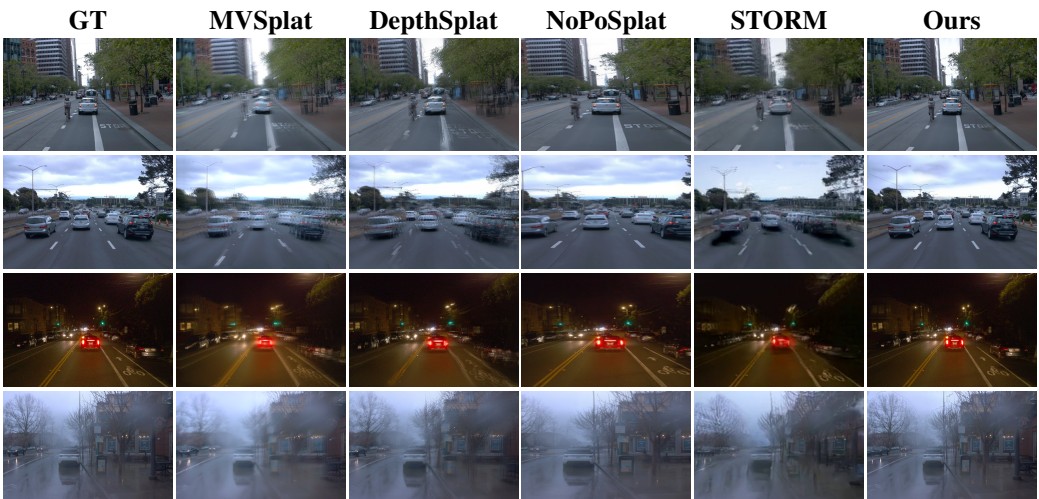

Figure 3: **Qualitative comparison of different methods on Waymo dataset.**

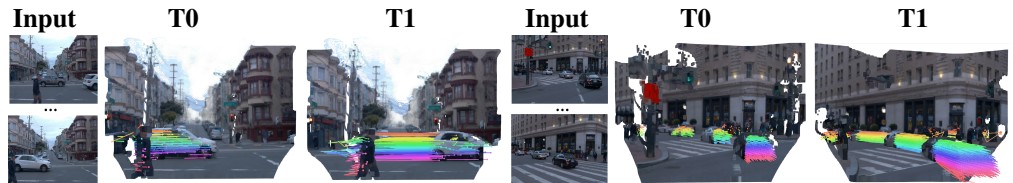

Figure 4: **3D Tracking Visualization**. Points with the same color correspond across frames.

PVG Chen et al. (2023), and DeformableGS Yang et al. (2024b), as well as feedforward methods including LGM Tang et al. (2024), GS-LRM Zhang et al. (2024b), MVSplat Chen et al. (2024b), NoPoSplat Ye et al. (2024), DepthSplat Xu et al. (2025), and STORM Yang et al. (2024a). For 3D tracking, we compare with NSFP Li et al. (2021), NSFP++ Najibi et al. (2022), and STORM.

**Novel View Synthesis**  To evaluate both reconstruction on input frames and novel view synthesis on interpolated frames, we use 8-frame sequences from 202 scenes as the test set, taking 4 frames as input to predict the intermediate frames and evaluating performance on all 8 frames. Tab. 1 reports results on the Waymo dataset, where our method outperforms all baselines in PSNR and SSIM. Fig. 3 further provides qualitative comparisons with feedforward approaches. While MVSplat, No-PoSplat, and DepthSplat fail to capture dynamic components due to their limited architectural design, STORM also struggles to accurately model object motion. In contrast, our model successfully distinguishes moving objects from the static background and produces faithful reconstructions. To further assess generalization, we evaluate on nuScenes and Argoverse2 (Tab. 2) under both zero-shot and training-from-scratch settings. In both cases, our model achieves the best performance, demonstrating strong adaptability and robustness across diverse driving scenarios. We credit this generalization ability to the pose-free design, which helps reduce domain gaps between datasets..

**3D Motion Estimation**  We evaluate motion estimation on the Waymo Scene Flow dataset. As summarized in Tab. 5, our approach consistently outperforms prior arts across all metrics. These results demonstrate that reliable correspondences can be effectively learned through the rendering loss. Fig. 4 presents qualitative 3D tracking results at timestamps $T_0$ and $T_1$. We render the 3D motion vectors onto the point clouds, where color-coded tracks highlight consistent correspondences across frames. These results show our method accurately recovers dynamic trajectories for vehicles and pedestrians, validating the motion estimation model.

### 4.3 ABLATION STUDY AND APPLICATION

**Number of Input Views**  One advantage of our method over previous feedforward approaches is its support for a flexible number of input views while maintaining strong performance on long se-

| Method | nuScenes | | | Argoverse2 | | |
|---|---|---|---|---|---|---|
| | PSNR ↑ | SSIM ↑ | LPIPS ↓ | PSNR ↑ | SSIM ↑ | LPIPS ↓ |
| **Zero-shot** | | | | | | |
| MVSplat Chen et al. (2024b) | 17.84 | 0.563 | 0.451 | 18.67 | 0.647 | 0.304 |
| NoPoSplat Ye et al. (2024) | 19.75 | 0.545 | 0.394 | 22.00 | 0.646 | 0.237 |
| DepthSplat Xu et al. (2025) | 19.52 | 0.601 | 0.376 | 22.05 | 0.636 | 0.280 |
| STORM Yang et al. (2024a) | 17.77 | 0.669 | 0.394 | 20.83 | 0.542 | 0.326 |
| Ours | **25.31** | **0.794** | **0.152** | **26.34** | **0.812** | **0.155** |
| **Trained** | | | | | | |
| STORM Yang et al. (2024a) | 24.54 | 0.784 | 0.267 | 24.97 | 0.791 | 0.240 |
| Ours | **26.63** | **0.813** | **0.122** | **26.96** | **0.831** | **0.118** |

Table 2: **Quantitative Comparison under Trained and Zero-Shot Settings on nuScenes and Argoverse2 datasets.** Our Waymo-trained model shows generalization ability in zero-shot NVS on other datasets, with training on each target dataset further boosting performance.

| Method | #Views Input | Reconstruction | | | NVS | | |
|---|---|---|---|---|---|---|---|
| | | PSNR ↑ | SSIM ↑ | D-RMSE ↓ | PSNR ↑ | SSIM ↑ | D-RMSE ↓ |
| STORM Yang et al. (2024a) | 4 | 26.55 | 0.851 | 6.139 | 26.05 | 0.819 | 5.914 |
| Ours | | **30.54** | **0.884** | **3.673** | **27.41** | **0.846** | **5.565** |
| STORM Yang et al. (2024a) | 8 | 25.11 | 0.807 | 6.054 | 25.44 | 0.807 | **5.470** |
| Ours | | **31.41** | **0.895** | **3.508** | **27.74** | **0.858** | 5.703 |
| STORM Yang et al. (2024a) | 16 | 23.69 | 0.765 | 5.700 | 22.98 | 0.723 | **5.836** |
| Ours | | **30.66** | **0.887** | **3.550** | **28.14** | **0.885** | 5.933 |

Table 3: **Ablation study on the number of input views.** Reconstruction performance shows low sensitivity to the number of input frames, demonstrating robustness with sparse inputs.

quences. Tab. 4 reports both reconstruction results on input frames and novel view synthesis (NVS) results on interpolated frames with 4, 8, and 16 input views. Our approach not only consistently outperforms STORM across all evaluation metrics, but also remains stable as the number of views increases. In contrast, STORM exhibits a sharp drop in PSNR and SSIM as views increase, demonstrating our superior scalability and robustness under varying input settings.

**Lifespan Parameter** We conduct an ablation study to assess the contributions of different model components, summarized in Tab. 4 and Fig. 5. Removing the lifespan parameter causes a significant performance drop, with PSNR decreasing from 27.41 to 24.21. This occurs because lifespan is crucial for capturing the dynamic appearance of static objects, such as lighting changes over time (see Fig. 5, row 3). Without it, Gaussians fail to model these subtle variations, resulting in unstable reconstructions and degraded perceptual quality.

**Diffusion Refinement** To evaluate the effectiveness of our diffusion refinement module, we conduct an ablation study with and without it, as summarized in Tab. 4 and visualized in Fig. 5. Quantitatively, the refinement improves PSNR and SSIM, confirming its role in enhancing rendering quality. Qualitatively, without refinement the reconstructions show noticeable artifacts and loss of fine details, particularly in the sky and dynamic objects. In contrast, incorporating diffusion refinement effectively removes these artifacts and produces more realistic renderings.

**Scene Editing** We further demonstrate the scene editing capability of our feedforward 4D reconstruction model. Since the method directly generates 3D Gaussian representations and decomposes them into dynamic and static components, we can flexibly manipulate the reconstructed scene by adding, removing, or repositioning dynamic objects. As illustrated in Fig. 6, the first row shows examples where cars are removed or shifted by modifying the dynamic Gaussians, while the second row demonstrates the seamless composition of novel vehicles and a cyclist by integrating the static scene with dynamic Gaussians reconstructed from other scenes. This editing process highlights the

Table 4: **Ablation study.** Removing lifespan parameters or the diffusion refinement model decreases performance.

| Method | PSNR ↑ | SSIM ↑ | LPIPS ↓ |
|---|---|---|---|
| w/o lifespan | 24.21 | 0.774 | 0.169 |
| w/o diffusion | 27.32 | 0.844 | **0.108** |
| Ours | **27.41** | **0.846** | 0.109 |

Table 5: **Quantitative results on 3D motion estimation.** We achieve state-of-the-art performance in 3D tracking on the Waymo Open Dataset.

| Method | EPE3D (m) ↓ | $Acc_5$ (%) ↑ | $Acc_{10}$ (%) ↑ | $\theta$ (rad) ↓ |
|---|---|---|---|---|
| NSFP | 0.698 | 42.17 | 54.26 | 0.919 |
| NSFP++ | 0.711 | 53.10 | 63.02 | 0.989 |
| STORM | 0.276 | 81.12 | 85.61 | 0.658 |
| Ours | **0.183** | **85.42** | **90.42** | **0.328** |

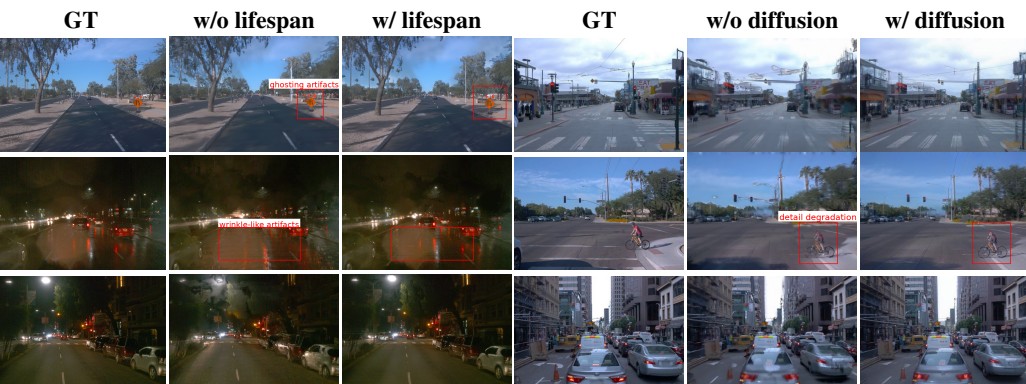

Figure 5: **Ablation study**. Removing the lifespan parameter hinders the capture of changing appearance of static scene, while the diffusion refinement reduces artifacts and improves rendering quality.

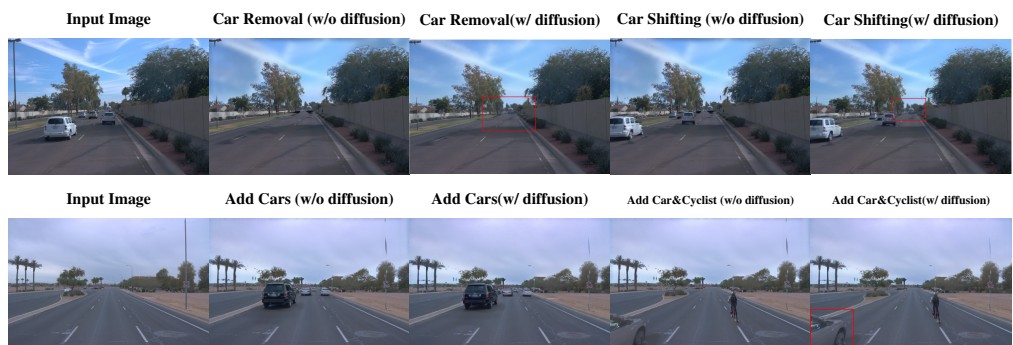

Figure 6: **Scene editing results**. Cars can be removed or shifted (row 1), and novel vehicles/cyclists inserted from other scenes (row 2). Diffusion refinement fixes artifacts such as holes (red box).

advantage of our representation: scene elements can be recombined at the Gaussian level without retraining, enabling efficient modifications of complex driving scenarios. Moreover, the edited images can be further refined using our diffusion-based model, which effectively addresses issues such as holes or artifacts introduced during Gaussian manipulation, as highlighted in the red box.

# 5 CONCLUSIONS

We proposed a pose-free, feedforward framework for 4D scene reconstruction from unposed images. Our method unifies camera estimation, 3D Gaussian reconstruction, and 3D motion prediction, while a diffusion-based refinement module improves fidelity under sparse inputs. Experiments on large-scale datasets show SOTA performance with fast speed. Despite these strengths, some limitations remain. In particular, failure cases can occur when dynamic masks are inaccurate or when tracking fails under heavily occluded motion. Future work will focus on improving dynamic modeling and enhancing tracking robustness to better handle complex dynamic scenes.

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

## A  IMPLEMENTATION DETAILS

### APPENDIX A.1 — DATA PREPROCESSING

We construct dynamic masks based on the LiDAR-based 3D bounding-box annotations of the Waymo Open DatasetSun et al. (2020), which include tracking identifiers. Specifically, we first transform the center of each 3D box from the ego-vehicle coordinate frame to the global coordinate system, and aggregate per-object temporal trajectories using the provided timestamps. Object velocities are then estimated, and category-specific thresholds are applied to determine dynamic instances (pedestrians: $> 0.2$ m/s; vehicles: $> 0.5$ m/s). For objects identified as dynamic, we project their 3D bounding boxes onto the corresponding camera planes to obtain accurate 2D bounding boxes for each frame. These 2D boxes are subsequently used as prompts for an *off-the-shelf* instance segmentation model Ravi et al. (2024), combined with temporal information to propagate masks and obtain temporally consistent, object-level instance masks. Static background and sky masks are produced using a semantic segmentation model Xie et al. (2021), whose semantic outputs are further employed in downstream scene understanding tasks.

### APPENDIX A.2 — BASELINE IMPLEMENTATIONS

**Baseline selection.**  For all per-scene reconstruction methods shown in Table 1, we selected ten representative baselines. Among them, PVGChen et al. (2023), DeformableGSYang et al. (2024b), 3DGSKerbl et al. (2023), and STORMYang et al. (2024a) are reconstruction methods implemented on the Waymo Open DatasetSun et al. (2020); MVSplatChen et al. (2024b) and DepthSplatXu et al. (2025) are feedforward inference networks with strong empirical performance; NoPoSplatYe et al. (2024) is a *pose-free* inference framework that does not require camera pose inputs. In addition, we include EmerNeRFYang et al. (2023), LGMTang et al. (2024), and GS-LRMZhang et al. (2024b) from STORM's reproduced results. For some baselines, reproduced outputs reported by STORM are used directly.

**Task setup.**  The task in Tab.1 and Fig.3 is short-sequence reconstruction and prediction: given four input frames (id = 0, 2, 4, 6), reconstruct eight frames (id = 0–7). This represents a basic scene reconstruction and novel view synthesis (NVS) task. Across the ten experiments, methods that require iterative fitting are uniformly trained/fitted for 5,000 iterations; feedforward reconstruction methods are evaluated without this iterative training constraint. All evaluations are performed on the Scene Flow validation split of the Waymo Open DatasetSun et al. (2020), which contains 202 scenes.

**Sky mask and depth metrics.**  Some methods do not explicitly reconstruct the sky; therefore depth-related metrics are computed only over non-sky regions. Sky masks are obtained from Waymo's LiDAR data and filtered to ensure high confidence. For depth evaluation, methods without camera pose input, including NoPoSplat and ours, predict only relative depth, whose scale and offset may differ from ground truth. To ensure fair comparison, we perform a linear alignment of the predictions before computing the error, and then report the aligned depth RMSE (D-RMSE) within the valid mask regions.

**Computation.**  During the training phase, the model was trained on the Waymo Open DatasetSun et al. (2020) using an eight-card H200 GPU configuration. The training process was completed in approximately 24 hours, with convergence achieved at around 5,000 iterations, as indicated by the stabilization of loss values and performance metrics on the validation set. In the experimental phase, to ensure direct comparability with STORMYang et al. (2024a) and its reproduced baseline methods, all evaluations in this study were exclusively conducted on NVIDIA A100 GPUs. This hardware alignment guarantees a consistent and fair comparison of computational performance and results across different models.

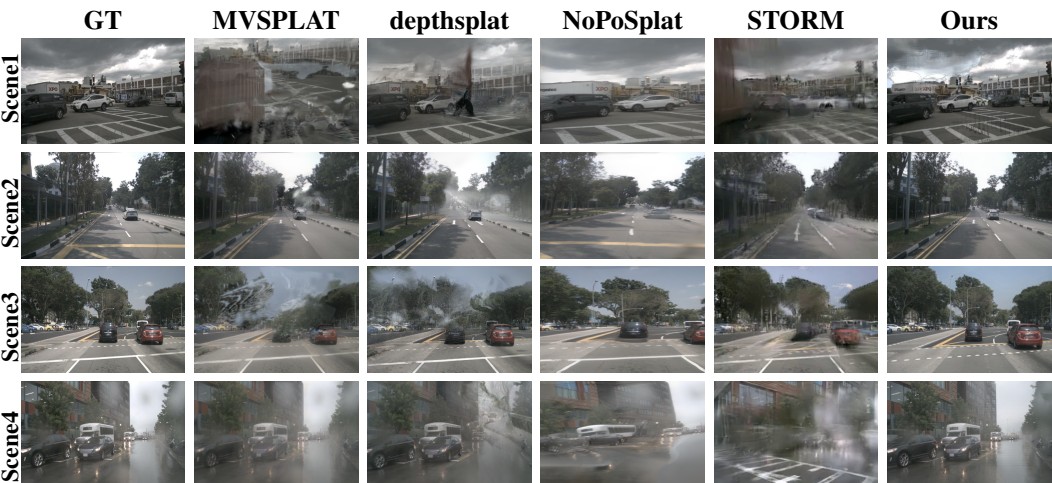

Figure 7: Zero-shot experiment on nuScenes and Argoverse2 datasets.

# B ADDITIONAL EXPERIMENTAL RESULTS

### APPENDIX B.1 — COMPARISON ON ADDTIONAL DATASETS

We evaluate the generalization of our model on the public nuScenes Caesar et al. (2020) and Argoverse2Wilson et al. (2023) datasets. To provide a systematic comparison of cross-domain and in-domain performance, we design two complementary experiments: (1) **zero-shot evaluation**, where the model is trained on the Waymo Open DatasetSun et al. (2020) and tested directly on nuScenes/Argoverse2 to assess cross-domain generalization; (2) **target-domain training evaluation**, where the model is trained and evaluated independently on nuScenes and Argoverse2 to measure upper-bound performance within each target domain.

Sampling and split details are as follows. For nuScenes (v1.0), the dataset contains approximately 1,000 driving scenes, each lasting roughly 20 s, with camera sampling at 12 Hz. We randomly sample 600 scenes for this study, using the first 500 scenes for training and the remaining 100 scenes for testing. For the Argoverse2 Sensor Dataset, which comprises roughly 1,000 annotated driving sequences and provides multi-modal sensor observations, the camera trigger frequency is approximately 20 Hz; we likewise select 600 sequences, with 500 used for training and 100 for evaluation.

To ensure comparability across datasets, all experiments adopt the same preprocessing and training configuration used for Waymo: camera images are uniformly downsampled/resampled to $518 \times 518$, and data augmentation, optimizer settings, and training schedules are kept consistent. Models typically converge after roughly 1,000 epochs. Selected zero-shot inference examples on nuScenes and Argoverse2 are shown in Fig. 7.

### APPENDIX B.2 — MORE QUALITATIVE RESULTS

Fig. 8 presents additional qualitative results of our method, showing full-image renderings, dynamic object renderings, and the predicted dynamic masks. Our approach effectively separates dynamic elements, such as vehicles and pedestrians, from the static background across diverse urban driving scenarios. The dynamic renderings align closely with ground-truth object locations, and the dynamic maps provide accurate object masks, demonstrating the effectiveness of our dynamic scene modeling and motion decomposition.

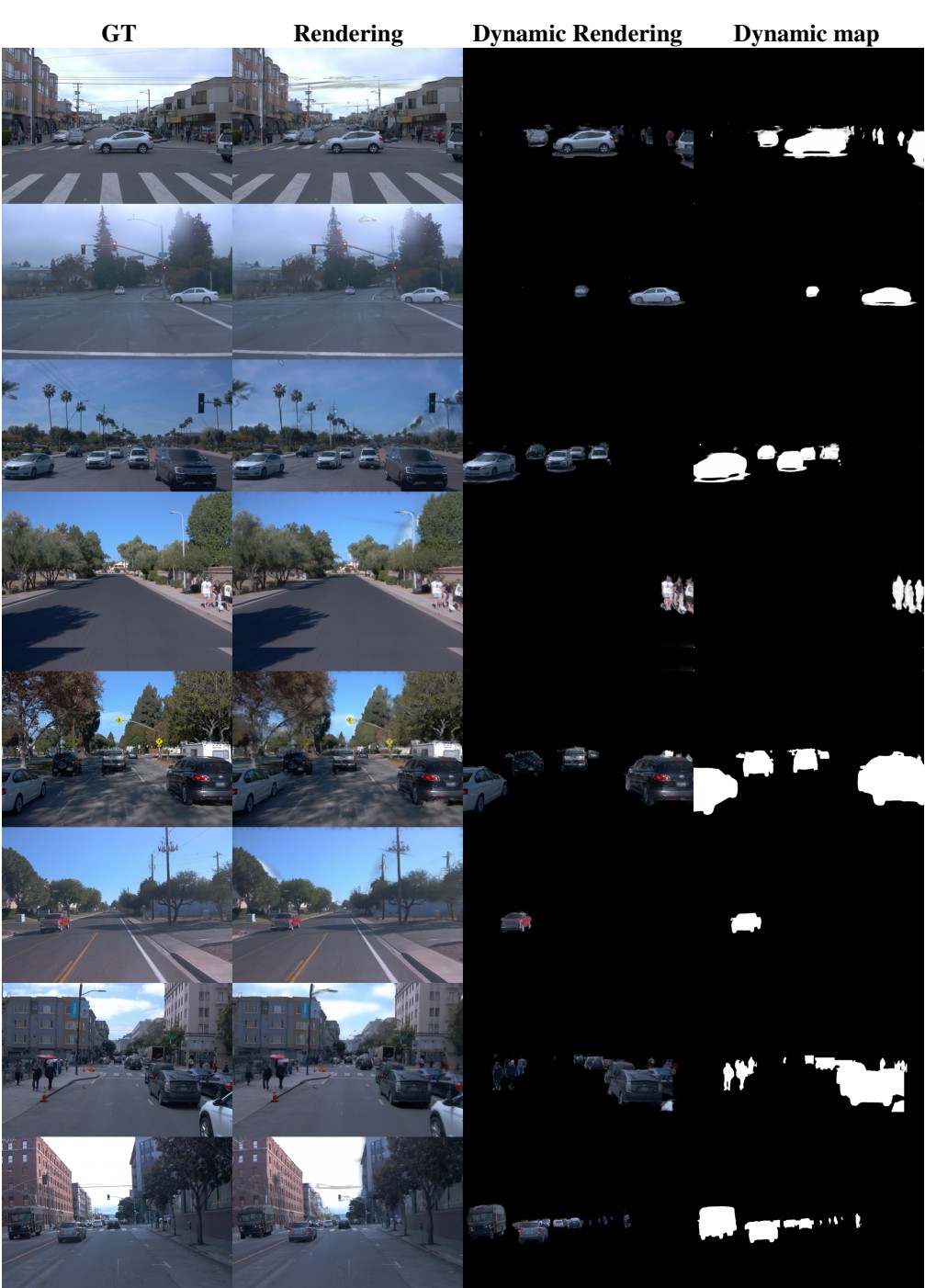

Figure 8: More Qualitative Results.

