# OpenReview forum: "Feedforward 4D Reconstruction for Dynamic Driving Scenes using Unposed Images"
_ICLR.cc/2026/Conference — ICLR 2026 Conference Withdrawn Submission_

### Official Review · Reviewer_enxr · 2025-10-23

**Soundness:** 3
**Presentation:** 3
**Contribution:** 2
**Rating:** 4
**Confidence:** 5

**Summary:**

This paper proposes a pose-free, feedforward 4D scene reconstruction framework to address the limitations of existing methods (slow per-scene optimization, reliance on annotations/camera calibration) for autonomous vehicle training. It jointly infers camera parameters, dynamic Gaussian representations, and 3D motion directly from sparse, unposed images—unlike prior feedforward approaches, it supports an arbitrary number of input views for long-sequence modeling. Key designs include disentangling dynamic objects via motion estimation (aggregated into 3DGS) and a diffusion-based module to reduce flow artifacts for better novel view synthesis. Trained on Waymo and tested on nuScenes/Argoverse2, it achieves superior cross-dataset generalization (thanks to pose-free design reducing biases) and supports instance editing + high-fidelity synthesis, serving as a scalable foundation for autonomous driving simulation.

**Strengths:**

1. Interesting questions and clear definitions of the questions.
2. Clear expression of methods and presentation of algorithms.
3. Reasonable ablation experiments.

**Weaknesses:**

I will convert the feedback into formal, academic English, ensuring each point is clear and targeted, while maintaining the professional tone suitable for paper review comments.

1. The algorithm's innovation is limited. Diffusion model-based restoration and reconstruction follow a conventional scene-centric mindset. Additionally, there is no comparison with the latest methods, such as novel view synthesis approaches like ReconDreamer, FreeVS, and Dist-4D.
2. There is a lack of visualizations for novel view synthesis. It is recommended to include visualizations of translations by 1m, 2m, and 4m to demonstrate the method's performance.
3. The image quality presented in the paper is insufficient for real-world applications, and there is a significant gap compared to the novel view synthesis results of ReconDreamer.
4. The reconstruction results shown in the demo appear to be of poor quality, and there remains a large performance gap relative to the demos provided by OmniRe.
5. The resolution discussed in the paper is relatively low, making it difficult to apply the method in practical scenarios.

**Questions:**

Does the thesis support multiple views? There are no relevant visualizations in the paper. The official demo provided by STORM doesn't seem to perform as poorly as described in the paper. Why is the baseline effect in the paper so poor?

---

### Official Review · Reviewer_baEa · 2025-10-30

**Soundness:** 2
**Presentation:** 3
**Contribution:** 2
**Rating:** 4
**Confidence:** 5

**Summary:**

This paper introduces a pose-free, feedforward framework for 4D dynamic scene reconstruction from unposed images. The method jointly estimates camera parameters, 3D Gaussian representations, and 3D motion in a single forward pass, without relying on per-scene optimization or external annotations. It incorporates a diffusion-based refinement module to enhance rendering quality, and demonstrates strong generalization across multiple autonomous driving datasets.

**Strengths:**

1. Feedforward Design for Speed and Generalization: The framework is entirely feedforward, enabling fast 4D scene reconstruction from unposed images in a single pass (0.39s). This design eliminates the need for slow per-scene optimization and allows the model to generalize effectively across diverse datasets like Waymo, nuScenes, and Argoverse2.
2. Single-Step Diffusion for Refinement: A key innovation is the integration of a single-step diffusion model as a rendering refinement module. This component effectively mitigates artifacts from sparse inputs and motion interpolation, significantly enhancing the fidelity and realism of the synthesized novel views without complex, multi-step denoising.
3. Explicit Dynamic-Static Decomposition with Motion Modeling: The method explicitly decomposes the scene into static and dynamic components. By estimating a 3D motion field for dynamic objects, it enables temporally consistent fusion and accurate interpolation of Gaussians for rendering at arbitrary timestamps, which also facilitates instance-level scene editing.

**Weaknesses:**

1. The experimental setup in Table 1: Quantitative comparison on the Waymo dataset differs from the STORM method. STORM is multi-view and uses an input frame interval of 5, while your method is single-view with an input frame interval of 1, which lacks fairness.
2. The design of lifespan is originally intended to better model the dynamic appearance of static objects, such as traffic lights. However, under the supervision of the render loss, this design may prevent static objects from truly aggregating. Specifically, the network may tend to learn a smaller lifespan, causing static Gaussians to appear only in the current/adjacent frames to achieve better rendering performance. This is also reflected in your ablation study on lifespan.
3. According to the ablation study, the performance improvement brought by the one-step diffusion design is very limited, which contradicts the qualitative results shown in your provided figures.

**Questions:**

See weaknesses

---

### Official Review · Reviewer_sVfb · 2025-10-31

**Soundness:** 3
**Presentation:** 3
**Contribution:** 2
**Rating:** 4
**Confidence:** 4

**Summary:**

This paper introduces a feedforward framework for 4D scene reconstruction that simultaneously estimates camera parameters and 3D scene representations from unposed images in a single pass. Built on a vision-transformer backbone, the model predicts per-frame 3D Gaussian representations and motion to capture dynamic scenes without requiring camera calibration or instance-level annotations. To handle motion-induced artifacts, a diffusion-based refinement module enhances reconstruction quality and novel view synthesis under sparse-view settings.

**Strengths:**

- The proposed method significantly outperforms state-of-the-art approaches, such as STORM, by a considerable margin.
- The proposed method is **pose-free**, which represents a key advantage and contributes to its robustness.

**Weaknesses:**

- The mathematical notation should be consistent. For example, w^{t_i} in Equation (6) differs from w_{t_i} in line 238.
- The plot on the right side of **Figure 1** may be misleading. Specifically, the label ‘m’ on the top x-axis should be replaced with ‘min’. In addition, the alignment between the top and bottom x-axes is inconsistent: 10 seconds approximately correspond to 1 FPS, while 1 second corresponds to about 10 FPS. Please clarify this relationship.
- The inference time of STORM is reported as **0.18 sec**, whereas the replicated version takes **0.50 sec**. Please explain the reason for this discrepancy. If STORM indeed runs at 0.18 sec, the proposed method’s inference time would be roughly double.

**Questions:**

- Are the STORM results in Tables 2 and 3 reproduced by the authors, or are they taken directly from the original paper?
- In **Figure 2**, which component corresponds to f_{\theta}? Do the authors use pretrained parameters from VGGT (Wang *et al.*, 2025a) for the feedforward model f_{\theta}?

---

### Official Review · Reviewer_h1Bi · 2025-11-05

**Soundness:** 3
**Presentation:** 2
**Contribution:** 2
**Rating:** 4
**Confidence:** 4

**Summary:**

This paper presents a feedforward framework for dynamic 4D scene reconstruction from unposed images. The model jointly predicts camera parameters, per-frame 3D Gaussian splatting (3DGS) representations, dynamic maps, and 3D motion in a single forward pass. A diffusion-based rendering refinement module further improves visual fidelity and reduces artifacts. Experiments on Waymo, nuScenes, and Argoverse2 demonstrate strong quantitative and qualitative results, outperforming prior methods while maintaining fast inference speed.

**Strengths:**

* The proposed method and its individual components are clearly motivated.
* The work introduces a single unified model that jointly predicts camera poses, 3D scene representations, and motion, making it versatile and promising for scalable simulation and reconstruction tasks.
* The experiments are comprehensive, covering multiple driving datasets and demonstrating consistently strong quantitative and qualitative performance.

**Weaknesses:**

* While the support for unposed images and variable input numbers is valuable, these improvements largely stem from the underlying VGGT architecture, and the refinement module is adapted from DifFix3D+. As a result, the paper feels more like a thoughtful combination of existing techniques rather than a fundamentally new contribution.
* The dynamic mask generation pipeline depends on off-the-shelf segmentation models and LiDAR-based annotations during preprocessing, which weakens the claim of being fully “annotation-free.”
* Although the paper includes some scene editing examples, they are limited to object addition and removal. It remains unclear how the method performs under more challenging conditions, such as extreme ego-motion or strong scene dynamics.
* While the diffusion-based refinement improves visual realism, it adds extra computational overhead and may introduce temporal inconsistency. How are these issues mitigated or evaluated?

**Questions:**

* The proposed method does not appear to be limited to driving scenes and could naturally extend to general 4D dynamic scenarios. Have the authors explored results on other datasets or domains to demonstrate this generalization?
* The proposed lifespan Gaussian shares similarities with the Periodic Vibration Gaussian (PVG). It would be helpful to better contextualize this design by referencing or comparing to PVG and related approaches.

---

### Note · Authors · 2025-11-14

I have read and agree with the venue's withdrawal policy on behalf of myself and my co-authors.